# Do You Remember? Overcoming catastrophic forgetting for fake audio detection

## Abstract

Current fake audio detection algorithms achieve promising performances on most datasets. However, their performance may be significantly degraded when dealing with audio of a different dataset. The orthogonal weight modification to overcome catastrophic forgetting does not consider the similarity of some audio, including fake audio obtained by the same algorithm and genuine audio, on different datasets. To overcome this limitation, we propose a continual learning algorithm for fake audio detection to overcome catastrophic forgetting, called Regularized Adaptive Weight Modification (RAWM). Specifically, when fine-tuning a detection network, our approach adaptively computes the direction of weight modification according to the ratio of genuine utterances and fake utterances. The adaptive modification direction ensures the network can detect fake audio on the new dataset while preserving its knowledge of previous model, thus mitigating catastrophic forgetting. In addition, orthogonal weight modification of fake audios in the new dataset will skew the distribution of inferences on audio in the previous dataset with similar acoustic characteristics, so we introduce a regularization constraint to force the network to remember this distribution. We evaluate our approach across multiple datasets and obtain a significant performance improvement on cross-dataset experiments.

## 1 Introduction

Currently, fake audio detection has attracted increasing attention since the organization of a series of challenges, such as the ASVspoof challenge (Wu et al., 2015; Kinnunen et al., 2017; Todisco et al., 2019; Yamagishi et al., 2021) and the Audio Deep Synthesis Detection challenge (ADD) (Yi et al., 2022). In these competitions, deep neural networks have achieved great success. Currently, large-scale pre-trained models have gradually been applied to fake audio detection and achieved state-of-the-art results on several public fake audio detection datasets (Tak et al., 2022; Martín-Doñas & Álvarez, 2022; Lv et al., 2022; Wang & Yamagishi, 2021). Although fake audio detection achieves promising performance, it may be significantly degraded when dealing with audio of another dataset. The diversity of audio proposes a significant challenge to fake audio detection across datasets (Zhang et al., 2021b;a).

Some approaches have been proposed to improve detection performance across datasets. Monteiro et al. (2020) proposed an ensemble learning method to improve the detection ability of the model for unseen audio. Wang et al. (2020) designed a dual-adversarial domain adaptive network to learn more generalized features for different datasets. Both methods require some audio from the old dataset, but in some practical situations, it is almost impossible to obtain them. For instance, a pretrained model proposed by a company has been released to the public. It is unfeasible for the public to fine-tune it using the data belonging to the original company. Zhang et al. (2021b) proposed a data augmentation method to extract more robust features for detection across datasets, which is only suitable for the datasets with similar feature distribution. Ma et al. (2021) proposed the first continual learning method for fake audio detection, called Detecting Fake Without Forgetting (DFWF) inspired by Learning without Forgetting (LwF) (Li & Hoiem, 2017). The DFWF improves the detection performance by fine-tuning on the new dataset and overcomes catastrophic forgetting by introducing regularization. Although the above methods are evaluated as viable options, there are still some insufficient places, like the acquisition of previous data in the first two and deteriorating

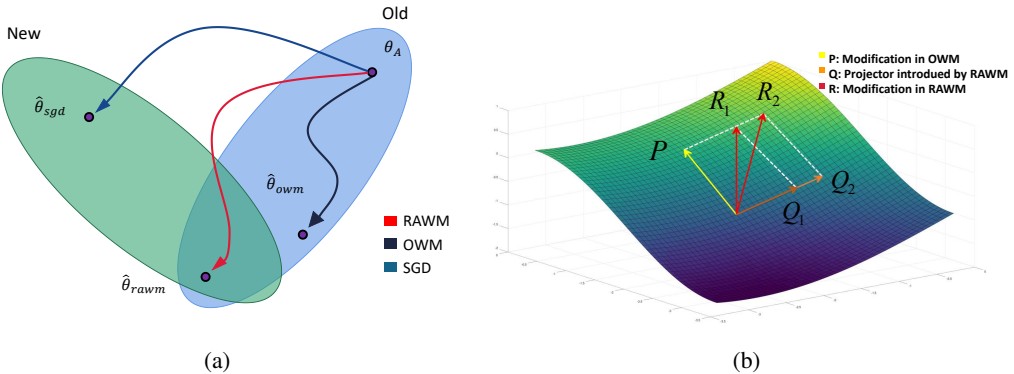

Figure 1: Schematic of SGD, OWM, and RAWM. **(a)**, With RAWM, the optimization process searches for configurations that lead to great performance on both old (blue area) and new (green area) datasets. A successful optimized configuration $\hat{\theta}_{rawm}$ stops inside the overlapping subspace. However, the configuration $\hat{\theta}_{sgd}$ obtained by SGD is optimized without considering forgetting, and the configuration $\hat{\theta}_{owm}$ obtained by orthogonal weight modification can not reach the overlapping region. **(b)**, the RAWM adaptively modifies weight direction by introducing a projector that is orthogonal to the projector $P$ proposed by OWM.

learning performance in the DFWF. This paper, however, aims to overcome catastrophic forgetting while exerting a positive influence on acquiring new knowledge without any previous samples.

As for fake audio detection, we have observed that most datasets are under clean conditions. Regarding these datasets, the genuine audio has a more similar feature distribution than the fake audio. Specifically, the variance of the feature distribution of genuine audio is smaller than that of fake audio (Ma et al., 2021). A few datasets, however, are collected under noisy conditions (Müller et al., 2022), which makes a great difference in their feature distributions of genuine audio (Ma et al., 2022). In this regard, if we modify the model weights as the orthogonal weight modification (OWM) method (Zeng et al., 2019) which introduces a new weight direction orthogonal to all previous data, most genuine audio can not be trained efficiently. The reason is that new data is supposed by the OWM to damage learned knowledge for its different feature distribution but it is unreasonable for fake audio detection. It is more efficient for most genuine audio to be trained with the same direction modification because of their similar feature distributions. To address these issues, we propose a continual learning approach, named Regularized Adaptive Weight Modification (RAWM). Because genuine audio has more similar feature distribution, it is reasonable to modify model weights in the same direction as the old one. Specifically, if the proportion of fake audio is larger, the modified direction is closer to the orthogonal projector of the subspace spanned by all previous input; if the proportion of genuine audio is larger, the modification is closer to the previous input subspace. However, when the feature distributions of old and new genuine audio are quite different, the effect of the above method is not obvious. We address this issue by introducing a regularization constraint. This constraint forces the model to remember the feature distribution without requiring prior knowledge. In addition, compared with the experience-replay-based method in continuous learning, RAWM does not require previous data, which makes this method suitable in most situations. Finally, the optimization process of RAWM is compared with that of the Stochastic Gradient Descent search (SGD) and OWM in Figure 1a.

**Contributions:** We propose a regularized adaptive weight modification algorithm to overcome catastrophic forgetting for fake audio detection. There are two essential modules in our method: adaptive weight modification (AWM) and regularization. The former AWM is proposed for continual learning in most situations where genuine audio has similar feature distribution and the latter regularization is introduced to ease the problem that genuine audio may have different feature distribution in a few cases. The experimental results show that our proposed method outperforms several continual learning methods in acquiring new knowledge and overcoming forgetting, including Elastic Weight Consolidation (EWC) (Kirkpatrick et al., 2017), LwF, OWM, and DFWF. The code will be publicly available in the foreseeable future.

## 2 RELATED WORK

In continual learning, overcoming catastrophic forgetting methods can be divided into the following categories. The regularization methods perform a regularization on the objection function or regulate important weights that are essential for previous tasks (Kinnunen et al., 2017; Zenke et al., 2017b; Aljundi et al., 2018; 2019; Mallya & Lazebnik, 2018; Serra et al., 2018). The dynamic architecture methods reserve their previous knowledge by introducing additional layers or nodes and grow model architecture (Rusu et al., 2016; Schwarz et al., 2018); (Yoon et al., 2017). The memory-based methods remember their previous data to prevent gradient updates from damage on their learned knowledge. (Lopez-Paz & Ranzato, 2017; Castro et al., 2018; Wu et al., 2019; Lee et al., 2019). The natural gradient descent methods approximate the Fisher information matrix in EWC using the generalized Gauss-Newton method to fast gradient descent (Tseran et al., 2018; Chen et al., 2019).

Although the mainstream continual learning methods, such as the EWC, LwF and OWM, have achieved great success in many fields including image classification (Zeng et al., 2019; Kirkpatrick et al., 2017), object detection (Perez-Rua et al., 2020), semantic segmentation (Cermelli et al., 2020), lifelong language learning (de Masson D'Autume et al., 2019) and sentence representation (Liu et al., 2019). However, the approximation of regularization methods will produce error accumulation in continual learning (Zenke et al., 2017a; Huszár, 2017; Ma et al., 2021). In contrast, our proposed method only needs the current inputs and some hyperparameters of the last task, which leads to a better performance our method achieves than others in error accumulation. Compared with the DFWF, we relax its regularized constraint and introduce a direction modification.

## 3 BACKGROUND

### 3.1 ORTHOGONAL WEIGHT MODIFICATION

The OWM algorithm overcomes catastrophic forgetting by modifying the direction of weights on the new task. The modified direction $\boldsymbol{P}$, which is a square matrix, is orthogonal to the subspace spanned by all inputs of the previous task. The orthogonal projector is constructed by an iterative method similar to the RLS algorithm (Shah et al., 1992), which hardly requires any previous samples.

We consider a feed-forward network consisting of $L+1$ layers, indexed by $l = 0, 1, \cdots, L$ with the same activation function $g(\cdot)$. The $\overline{\mathbf{x}}_l(i, j) \in \mathbb{R}^s$ represents the output of the $l$th layer in response to the mean of the $i$th batch inputs on $j$th dataset, and the $\overline{\mathbf{x}}_l(i, j)^T$ is the transpose matrix of the $\overline{\mathbf{x}}_l(i, j)$. The modified direction $\boldsymbol{P}$ can be calculated as:

$$\boldsymbol{P}_l(i, j) = \boldsymbol{P}_l(i-1, j) - \mathbf{k}_l(i, j)\overline{\mathbf{x}}_{l-1}(i, j)^T \boldsymbol{P}_l(i-1, j)$$
$$\mathbf{k}_l(i, j) = \frac{\boldsymbol{P}_l(i-1, j)\overline{\mathbf{x}}_{l-1}(i, j)}{\alpha + \overline{\mathbf{x}}_{l-1}(i, j)^T \boldsymbol{P}_l(i-1, j)\overline{\mathbf{x}}_{l-1}(i, j)} \tag{1}$$

where $\alpha$ is a hyperparameter decaying with the number of tasks. This iterative algorithm only needs the current inputs and orthogonal projector for the last task, thus avoiding loading data from the previous task.

### 3.2 LEARNING WITHOUT FORGETTING

The LwF algorithm is inspired by the idea of model distillation, where old knowledge is viewed as a penalty term to regulate the new model representation similar to the old. Specifically, the model trained on old datasets is replicated into two models with the same parameters. The two models are named teacher and student models in the LwF. In process of training on new datasets, the parameters of the teacher model are frozen to produce its features as "soft labels". The student model is trained by the loss function as:

$$\boldsymbol{L}_{lwf} = \lambda_0 \boldsymbol{L}_{old}(\mathbf{y}_o, \hat{\mathbf{y}}_o) + \boldsymbol{L}_{new}(\mathbf{y}_n, \hat{\mathbf{y}}_n) \tag{2}$$

where $\lambda_0$ is a ratio coefficient representing the importance of learned knowledge; $\mathbf{y}_o$ is the "soft label" produced by the teacher model and $\mathbf{y}_n$ is the ground truth of new data; Both $\hat{y}_o$ and $\hat{y}_n$ are the softmax output of the student model. Both $\boldsymbol{L}_{old}$ and $\boldsymbol{L}_{new}$ are cross-entropy loss. The first encourages predictions $\hat{y}_n$ to be consistent with the ground truth $\mathbf{y}_n$ and the last regulates the output probabilities $\hat{y}_o$ to be close to the recorded output $\mathbf{y}_o$ from the teacher model.

# 4 PROPOSED METHOD

As for fake audio detection, we have observed some distinctive features in this regard. On most fake audio detection datasets, under the same acoustic conditions, feature distributions of genuine audio are relatively more concentrated and unified than the fake, which means the feature distribution of genuine audio has a smaller variance than that of fake audio (Ma et al., 2021; Yan et al., 2022). Apart from that, there are also a few datasets whose genuine audio has quite different feature distributions from others (Ma et al., 2022; Müller et al., 2022). For instance, genuine audio collected from noisy conditions may extremely skew their acoustic feature distribution. Although it seems not a norm in this field, we still are supposed to pay attention to these datasets since it was common in reality.

Based on this inference, we propose a continual learning method, named Regularized Adaptive Weight Modification (RAWM), to overcome catastrophic forgetting for fake audio detection. There are two essential parts in our method: adaptive direction modification (AWM) and regularization. The AWM, which is described in Sec. 4.1, is proposed for most situations where the feature distributions of genuine audio are very similar. By introducing an extra projector, which is a square matrix orthogonal to the projector proposed by the OWM, our method could adaptively modify weight direction forcing it closer to the previous inputs subspace. As for those genuine audio collected from noisy conditions, it is detrimental for learned knowledge to modify weight according to the rule we mentioned above, because their feature distribution is distinct from others. To address this issue, we introduce a regularization term to force the new distribution of inference to be similar to the old one, which is described in Sec. 4.2. Apart from that, we also present how our method regulates models by modifying weight direction under the restriction of regularization in Sec. 4.3 and show the process of our algorithm in Algorithm 1. On top of that, our method does not require any prior knowledge of both old and new datasets and replay of previous samples.

## 4.1 ADAPTIVE WEIGHT MODIFICATION

We start by introducing an adaptive modification of weight direction according to the ratio of genuine and fake audio in batch data, which is essential for sequence training on multi-datasets. We first consider a feed-forward network like that described in Sec. 3.1. Then, we introduce a square matrix $Q$ as a projector that is orthogonal to the $P$ proposed by the OWM algorithm. This orthogonal projector can be written as Eq 3:

$$Q = \beta[I - P(P^T P)^{-1} P] \tag{3}$$

where the projector $P$, which is orthogonal to the subspace spanned by all previous inputs, can be calculated as Eq 1 and $I$ is an identity matrix. The construction of the orthogonal projector $Q$ is mathematically sound (Haykin, 2002; Ben-Israel & Greville, 2003; Bengio & LeCun, 2007). To verify the modification direction according to the ratio of genuine audio and fake audio, we introduce the $\beta$ defined as:

$$\beta = \frac{N_g + 1}{N_f + 1} \tag{4}$$

in which $N_g$ and $N_f$ represent the number of genuine and fake audios in a batch, respectively. By adding one to both the numerator and denominator, $\beta$ can be calculated when all the batch audios are genuine. As illustrated in Eq 3, the norm of projector $Q$ is proportional to the ratio $\beta$. Our approach defines the modified direction $R$ of weights as:

$$R = P_N + mQ_N \tag{5}$$

$$P_N = \frac{P}{||P||}, \quad Q_N = \frac{Q}{||I - P(P^T P)^{-1}P||} \tag{6}$$

where m is a constant to constrain the norm of projector $Q$ to prevent gradient explosion or gradient vanishing in the backward process; $P_N$ and $Q_N$ are identity matrices normalized by $P$ and $Q$, respectively. Normalization is to prevent the case that the change of $\beta$ has little effect on the modified direction because of the large norm gap between $P$ and $Q$. In the back-propagate (BP) process, the direction of network weights is modified as:

$$W_l(i, j) = W_l(i-1, j) + \gamma(i, j)\Delta W_l^{BP}(i, j) \qquad when\ j = 1$$
$$W_l(i, j) = W_l(i-1, j) + \gamma(i, j)R_l(j-1)\Delta W_l^{BP}(i, j) \qquad when\ j > 1 \tag{7}$$

where $W_l(i, j) \in \mathbb{R}^{s \times v}$ represents the connection weights between the $l$th layer and the $(l+1)$th layer; $\gamma$ represents the learning rate of this network; $\Delta W_l^{BP}(i, j)$ represents the standard BP gradient; $R$ represents the modification projector in our method. In Eq 7, we can easily observe that we

---

**Algorithm 1** Regularized Adaptive Weight Modification

---

**Require:** Training data from different datasets, $\gamma$ (learning rate), m (constant hyperparameter), $T_{reg}$ (constant hyperparameter).

1: **for** every dataset $j$ **do**
2:     **repeat**
3:     **for** every batch $i$ **do**
4:         **repeat**
5:         **if** $j = 1$ **then**
6:             $\boldsymbol{W}_l(i,j) = \boldsymbol{W}_l(i-1,j) + \gamma(i,j)\Delta\boldsymbol{W}_l^{BP}(i,j)$    ▷ $\Delta\boldsymbol{W}_l^{BP}$ gradient by standard BP method
7:         **else**
8:             $\mathbf{k}(i,j) = \boldsymbol{P}_l(i-1)\overline{\mathbf{x}}_{l-1}(i,j)/[\alpha + \overline{\mathbf{x}}_{l-1}(i,j)^T\boldsymbol{P}_l(i-1,j)\overline{\mathbf{x}}_{l-1}(i,j)]$
9:             $\boldsymbol{P}_l(i,j) = \boldsymbol{P}_l(i-1,j) - \mathbf{k}(i,j)\overline{\mathbf{x}}_{l-1}(i,j)^T\boldsymbol{P}_l(i-1,j)$      ▷ $\overline{\mathbf{x}}_l$ output of the mean of inputs
10:           $\beta = \dfrac{N_g + 1}{N_f + 1}$                  ▷ $N_g$, $N_f$: genuine, fake audio number
11:           $\boldsymbol{Q} = \beta[\boldsymbol{I} - \boldsymbol{P}(\boldsymbol{P}^T\boldsymbol{P})^{-1}\boldsymbol{P}]$
12:           $\boldsymbol{P}_N = \dfrac{\boldsymbol{P}}{||\boldsymbol{P}||}$
13:           $\boldsymbol{Q}_N = \dfrac{\boldsymbol{Q}}{||\boldsymbol{I} - \boldsymbol{P}(\boldsymbol{P}^T\boldsymbol{P})^{-1}\boldsymbol{P}||}$
14:           $\boldsymbol{R} = \boldsymbol{P}_N + m\boldsymbol{Q}_N$
15:           $\hat{y}_o(i) = \dfrac{y_o(i)^{1/T_{reg}}}{\sum \mathbf{y}_o(i)^{1/T_{reg}}}$
16:           $\hat{y}_n(i) = \dfrac{y_n(i)^{1/T_{reg}}}{\sum \mathbf{y}_n(i)^{1/T_{reg}}}$
17:           $\Delta\boldsymbol{W}_{l_{reg}}^{BP} = -\nabla(\hat{\mathbf{y}}_o(i) \cdot \log\hat{\mathbf{y}}_n(i))$
18:           $\boldsymbol{W}_l(i,j) = \boldsymbol{W}_l(i-1,j) + \gamma(i,j)((1-\eta)\boldsymbol{R}_l(j-1)\Delta\boldsymbol{W}_l^{BP}(i,j) + \eta\Delta\boldsymbol{W}_{l_{reg}}^{BP}(i,j))$
19:         **end if**
20:         **until** loss plateaus
21:     **end for**
22: **end for**

---

modify weight direction adaptively by multiplying the BP gradient $\Delta\boldsymbol{W}_l^{BP}(i,j)$ with our projector $\boldsymbol{R}$ whose direction is varied according to the ratio of genuine and fake audio.

## 4.2   Regularization

There are a few datasets where genuine audio is collected from noisy conditions. In this case, it is unreasonable to use the above method directly. As for these utterances, we introduce an extra regularization forcing the model to remember the previous inference distribution.

We first replicate the pre-trained model into two models with the same parameters, one is the teacher model and the other one is the student model. The parameter of the teacher model is frozen in the process of training on the new dataset and the parameter of the student model is fine-tuned. Like the operation in the LwF, we view the softmax output $\mathbf{y}_o$ from the teacher model as "soft labels" and use the loss function to slash the distinction between the "soft labels" $\mathbf{y}_o$ and the softmax output $\mathbf{y}_n$ of the student model, thus forcing the student model to remember the learned knowledge. The loss function, which is a modified cross-entropy loss, can be written as:

$$\boldsymbol{L}_{reg}(\hat{\mathbf{y}}_o, \hat{\mathbf{y}}_n) = -\hat{\mathbf{y}}_o \cdot \log\hat{\mathbf{y}}_n \tag{8}$$

$$\hat{y}_o = \frac{y_o^{1/T_{reg}}}{\sum \mathbf{y}_o^{1/T_{reg}}}, \quad \hat{y}_n = \frac{y_n^{1/T_{reg}}}{\sum \mathbf{y}_n^{1/T_{reg}}} \tag{9}$$

where $T_{reg}$ is a constant hyperparameter. The $\mathbf{y_o}$, $\mathbf{y_n}$ are softmax outputs of teacher and student models, respectively; The $\hat{\mathbf{y}}$ is a normalized form of the $\mathbf{y}$; The $\hat{y}$ and $y$ are one item of $\hat{\mathbf{y}}$ and $\mathbf{y}$, respectively. The weight modification of this regularization $\Delta\boldsymbol{W}_{l_{reg}}^{BP}$ can be written as Eq 10.

$$\Delta\boldsymbol{W}_{l_{reg}}^{BP} = \nabla\boldsymbol{L}_{reg} \tag{10}$$

## 4.3   Regularized Adaptive Weight Modification

In brief, our continual learning method RAWM is proposed for fake audio detection by modifying weight direction according to the ratio of genuine and fake audio in a batch and eases the problem that a few utterances corrupted by noise may interfere with the direction modification by introducing a regularized restriction. Considering a continual learning situation, the BP process of regularized adaptive weight modification can be written as Eq 11.

$$
\begin{aligned}
\boldsymbol{W}_l(i,j) &= \boldsymbol{W}_l(i-1,j) + \gamma(i,j)\Delta\boldsymbol{W}_l^{BP}(i,j) & \text{when } j = 1 \\
\boldsymbol{W}_l(i,j) &= \boldsymbol{W}_l(i-1,j) + \gamma(i,j)((1-\eta)\boldsymbol{R}_l(j-1)\Delta\boldsymbol{W}_l^{BP}(i,j) + \eta\Delta\boldsymbol{W}_{l_{reg}}^{BP}(i,j)) & \text{when } j > 1
\end{aligned}
\tag{11}
$$

Compared with the Eq 7, our method introduces a regularization constraint to the adaptive weight modification. The importance of the regularization depends on the hyperparameter $\eta$ which is a coefficient measuring the attention degree of the old task.

## 5 EXPERIMENTS

### 5.1 DATASETS

We conduct our experiments on four fake audio datasets, including the ASVspoof2019LA ($\boldsymbol{S}$), ASVspoof2015 ($\boldsymbol{T_1}$), VCC2020 ($\boldsymbol{T_2}$), and In-the-Wild ($\boldsymbol{T_3}$). The models are firstly trained using the training set of the ASVspoof2019 and are fine-tuned on the training sets of the other three datasets. All of the experiments are evaluated using two or four evaluation sets in these datasets.

**ASVspoof 2019 LA Dataset** (Todisco et al., 2019) is the sub-challenge dataset (30 males and 37 females) containing three subsets: training, development, and evaluation. The training set and development share the same attack including four TTS and two VC algorithms. The bonafide audio is collected from the VCTK corpus (Veaux et al., 2017). The evaluation set contains totally different attacks.

**ASVspoof2015 dataset** (Wu et al., 2015) is an open-source standard dataset of genuine and synthetic speech in the ASVspoof2015 challenge. The genuine speech was recorded from 106 speakers (45 males and 61 females) with no significant channel or background noise effects. The spoofing speech is generated using a variety of speech synthesis and voice conversion algorithms.

**VCC2020 dataset** (Zhao et al., 2020) is collected from Voice Conversion Challenge 2020. This dataset contains two subsets: a set of genuine audio provided by organizers and a set of fake audio provided by participating teams. Different from the previous three datasets, VCC2020 is a multilingual fake audio dataset, including English, Finnish, German and Mandarin.

**In-the-Wild dataset** (Müller et al., 2022) contains a set of deep fake audio (and corresponding real audio) of 58 politicians and other public figures collected from publicly available sources, such as social networks and video streaming platforms. In total, 20.8 hours of genuine audio and 17.2 hours of fake audio were collected. On average, each speaker had 23 minutes of genuine audio and 18 minutes of fake audio.

We divide the genuine and fake audios of the VCC2020 dataset into four subsets. A quarter is used to build the evaluation set, a quarter to build the development set, and the rest to be used as the training set. The In-the-Wild dataset is divided in the same way as the VCC2020. The ASVspoof2015 is the most similar to the ASVspoof2019LA for their audios are collected from the same datasets or conversion algorithms. The audio of the In-the-Wild dataset is collected from the real world and the audio of the VCC2020 dataset is multilingual. The detailed statistics of the datasets are presented in Table 1. The Equal Error Rate (EER), which is widely used for fake audio detection and speaker verification, is applied to evaluate the experimental performance.

### 5.2 EXPERIMENTAL SETUP

**Fake audio detection Model**: We use the pre-trained model Wav2vec 2.0 (Baevski et al., 2020) as the feature extractor and the self-attention convolutional neural network (S-CNN) as the classifier. The parameters of Wav2vec 2.0 is loaded from the pre-train model XLSR-53 (Conneau et al., 2020). The classifier S-CNN contains three 1D-Convolution layers, one self-attention layer, and two full connection layers, according to the forward process. The input dimension of the first convolution layer is 256 and the hidden dimension of all convolution layers is 80. The kernel size and stride are

Table 1: Statistics of experimental datasets.

| Dataset | ASVSpoof2019 | | ASVSpoof2015 | | VCC2020 | | In-the-Wild | |
|---|---|---|---|---|---|---|---|---|
| | #Real | #Fake | #Real | #Fake | #Real | #Fake | #Real | #Fake |
| Train | 2,580 | 22,800 | 3,750 | 12,625 | 1,330 | 3,060 | 9,431 | 5,908 |
| Dev | 2,548 | 22,296 | 3,497 | 49,875 | 665 | 1,530 | 4,715 | 2,954 |
| Eval | 7,355 | 63,882 | 9,404 | 184,000 | 665 | 1,530 | 4,717 | 2,954 |

Table 2: The EER(%) of our baseline on multiple evaluation sets.

| Model | S | $\mathbf{T_1}$ | $\mathbf{T_2}$ | $\mathbf{T_3}$ |
|---|---|---|---|---|
| Baseline | 0.258 | 24.532 | 46.503 | 91.473 |

set to 5 and 1, respectively. The hidden dimension of all full connection layers is 80 and the output dimension of the last is 2.

**Training Details**: We fine-tune the model weights including the pre-trained model XLSR-53 and the classifier S-CNN. All of the parameters are trained by the Adam optimizer with a batch size of 2 and a learning rate $\gamma$ of 0.0001. The constant m and $T_{reg}$ in RAWM are set to 0.1 and 2, respectively. The $\alpha$ is initialized to 0.00001 for convolution layers, 0.0001 for the self-attention layer, and 0.1 for full connection layers. The norm in normalization of projector $\boldsymbol{P}$ and $\boldsymbol{Q}$ is the $\boldsymbol{L}^2$ norm. In addition, we present the results of training all datasets (Tain-on-All) that is considered to be the lower bound to all continual learning methods we mentioned (Parisi et al., 2019). All results are (re)produced by us and averaged over 7 runs with standard deviations.

## 5.3 BASELINE

We first train our model on the training set of the ASVspoof2019LA dataset. Table 2 shows the detection performance of our baseline on multiple evaluation sets which is very close to the state-of-the-art result in the same dataset (Nautsch et al., 2021). Although the model achieves promising performance on the ASVspoof2019LA, its detection accuracy degrades significantly on other datasets. In addition, our baseline achieves the lowest cross-datasets EER on the ASVspoof2015 dataset among three unseen datasets, which verifies that the fake audio generated by the same algorithms has similar feature distribution, while the feature distribution of fake audio generated by different algorithms is quite different. Apart from that, the results with different training steps are presented in Table 7 in the appendix.

## 5.4 THE EFFECTIVENESS OF THE $\eta$ FOR OUR METHOD

**Sequence training between two datasets**: We start by performing some experiments to evaluate the effectiveness of $\eta$ in RAWM, which represents the attention degree to learned knowledge. In Table 3, we can easily observe that the RAWM achieves great performance on both old and new datasets, especially in the experiment on $\mathbf{S} \rightarrow \mathbf{T_1}$. By comparing the results of three cross-datasets, we observe that when the new and old datasets have similar feature distribution (Table 3a), there is an improvement in the performance of both acquiring new knowledge and overcoming forgetting with the increasing of $\eta$ ($\eta < 1$); When the feature distribution of the new and previous datasets is different (Table 3b, Table 3c), it is the model when $\eta = 0.50$ that achieves the best result, which shows that regularization is also of benefit to performance on both learning and overcoming forgetting.

**Sequence training on four datasets**: We also present the results on multiple evaluation sets about different $\eta$ in Table 4a. It can be observed that our method slashes performance degradation when training across datasets. The RAWM achieves the lowest EER among the results when $\eta = 0.50$, which demonstrates that the same attention degree to both old and new datasets is the best choice for learning and overcoming forgetting. In addition, the results of dataset $\mathbf{T_3}$ show that smaller $\eta$ is more beneficial for models to acquire knowledge. Apart from that, the results of S, $\mathbf{T_1}$ and $\mathbf{T_2}$ show that the model with larger $\eta$ is more effective in overcoming forgetting.

Table 3: The EER(%) on evaluation sets of our method with different $\eta$. All experiments are trained using the training set in order to $\mathbf{S} \to \mathbf{T_k}$ and are evaluated using the evaluation set on $\mathbf{S}$ and $\mathbf{T_k}$

(a)

| $\eta$ | $\mathbf{S}$ | $\mathbf{T_1}$ |
|---|---|---|
| Baseline | 0.258 | 24.532 |
| 0.00 | 1.643 | 0.256 |
| 0.20 | 1.424 | 0.431 |
| 0.25 | 1.175 | 0.311 |
| 0.50 | 0.878 | 0.257 |
| **0.75** | **0.666** | **0.247** |
| 1.00 | 3.123 | 0.343 |

(b)

| $\eta$ | $\mathbf{S}$ | $\mathbf{T_2}$ |
|---|---|---|
| Baseline | 0.258 | 46.503 |
| 0.00 | 1.413 | 3.845 |
| 0.20 | 1.334 | 4.288 |
| 0.25 | 1.275 | 3.994 |
| **0.50** | **1.237** | **3.721** |
| 0.75 | 1.262 | 4.571 |
| 1.00 | 4.234 | 4.566 |

(c)

| $\eta$ | $\mathbf{S}$ | $\mathbf{T_3}$ |
|---|---|---|
| Baseline | 0.258 | 91.473 |
| 0.00 | 4.126 | 1.457 |
| 0.20 | 3.490 | 1.848 |
| 0.25 | 2.975 | 1.593 |
| **0.50** | **2.038** | **1.425** |
| 0.75 | 2.482 | 2.271 |
| 1.00 | 2.453 | 2.598 |

Table 4: The EER(%) on four evaluation sets. All experiments are trained using training set in order to $\mathbf{S} \to \mathbf{T_1} \to \mathbf{T_2} \to \mathbf{T_3}$ and are evaluated using evaluation sets.

(a) The EER(%) of the RAWM with different $\eta$.

| $\eta$ | $\mathbf{S}$ | $\mathbf{T_1}$ | $\mathbf{T_2}$ | $\mathbf{T_3}$ |
|---|---|---|---|---|
| Baseline | 0.258 | 24.532 | 46.503 | 91.473 |
| 0.00 | 1.845 | 1.127 | 3.916 | 1.410 |
| 0.20 | 1.724 | 1.003 | 4.120 | 1.367 |
| 0.25 | 1.699 | 0.945 | 4.017 | 1.529 |
| **0.50** | **1.508** | **0.641** | **3.850** | **1.163** |
| 0.75 | 1.636 | 0.873 | 3.975 | 2.454 |
| 1.00 | 2.714 | 1.621 | 3.875 | 2.325 |

(b) The EER(%) of our method compared with others.

| Method | $\mathbf{S}$ | $\mathbf{T_1}$ | $\mathbf{T_2}$ | $\mathbf{T_3}$ |
|---|---|---|---|---|
| Baseline | 0.258 | 24.532 | 46.503 | 91.473 |
| Train-on-All | 1.324 | 0.561 | 3.579 | 1.008 |
| Fine-tune | 7.068 | 2.841 | 5.674 | 2.543 |
| EWC | 5.569 | 3.444 | 4.510 | 2.129 |
| OWM | 4.083 | 2.167 | 4.480 | 2.472 |
| LwF | 2.714 | 1.621 | 3.875 | 2.325 |
| DFWF | 3.476 | 3.735 | 7.345 | 3.114 |
| **RAWM(Ours)** | **1.508** | **0.641** | **3.850** | **1.163** |

## 5.5 Ablation studies for our method

**Sequence training between two datasets**: In this section, we compare our proposed method with adaptive weight modification without regularization ($-$REG) and orthogonal weight modification without regularization ($-$AWM). Table 5 presents their EER on three evaluation sets. We observe that RAWM achieves similar EER to $-$REG on the new dataset, both of them are superior significantly to $-$AWM, which shows that the adaptive weight modification has a significant positive impact on acquiring knowledge, while regularization impacts little. As for overcoming forgetting, when the feature distribution of the new and old datasets is similar (Table 5a), the EER of the $-$REG on the old datasets is much lower than that of the $-$AWM and higher than that of the RAWM, which shows that the adaptive weight modification and regularization can significantly reduce the forgetting in this case. When the languages of the new and old datasets are different (Table 5b), the EER of RAWM in the old datasets is similar to that of the $-$REG and much lower than that of the $-$AWM, which also proves that the adaptive weight modification has a significant positive impact on overcoming forgetting. When the feature distribution of the new and old datasets is quite different (Table 5c), the EER of the $-$REG is similar to that of the $-$AWM and much higher than that of the RAWM, which shows that in this case, regularization is of great benefit to overcoming forgetting, while the effect of adaptive weight modification is not obvious.

**Sequence training on four datasets**: In this section, we present the results of the ablation study on four evaluation sets in Table 5d. We observe that the EER of $-$REG to $-$AWM degrades more obviously than that of RAWM to $-$REG on all evaluation sets, which indicates that adaptive weight modification has a more obvious benefit in learning and overcoming forgetting than regularization for sequence training on multiple datasets.

## 5.6 Comparison of our method with other methods

**Sequence training between two datasets**: We compare our method with several methods in Table 6. The EWC, LwF, and OWM as three mainstream continual learning methods achieve great success in many fields. The DFWF is the first continual learning method to overcome forgetting for fake audio detection. The results demonstrate that fine-tuning without modification (Fine-tune) forgets previous knowledge obviously. The forgetting of RAWM is one-tenth that of Fine-tune on Table 6a and the EER on the new dataset of RAWM is also half that of Fine-tune. We also observe that the

Table 5: The EER(%) on evaluation sets of the ablation studies. (a), (b) and (c) are trained using the training set in order to $S \rightarrow T_k$ and are evaluated using the evaluation set on $S$ and $T_k$; (d) is trained in order to $S \rightarrow T_1 \rightarrow T_2 \rightarrow T_3$ and are evaluated using evaluation sets.

(a)

| Method | S | $T_1$ |
|---|---|---|
| RAWM | **0.666** | **0.247** |
| −REG | 1.643 | 0.256 |
| −AWM | 2.448 | 0.500 |

(b)

| Method | S | $T_2$ |
|---|---|---|
| RAWM | **1.237** | **3.721** |
| −REG | 1.413 | 3.845 |
| −AWM | 3.086 | 5.432 |

(c)

| Method | S | $T_3$ |
|---|---|---|
| RAWM | **2.038** | **1.425** |
| −REG | 4.126 | 1.457 |
| −AWM | 4.857 | 2.663 |

(d)

| Method | S | $T_1$ | $T_2$ | $T_3$ |
|---|---|---|---|---|
| RAWM | **1.508** | **0.641** | **3.850** | **1.163** |
| −REG | 1.845 | 1.127 | 3.916 | 1.410 |
| −AWM | 4.083 | 2.167 | 4.480 | 2.472 |

Table 6: The EER(%) of our method compared with various methods. All experiments are trained using the training set in order to $S \rightarrow T_k$ and are evaluated using the evaluation set on $S$ and $T_k$

(a)

| Method | S | $T_1$ |
|---|---|---|
| Baseline | 0.258 | 24.532 |
| Train-on-All | 0.406 | 0.201 |
| Fine-tune | 7.324 | 0.510 |
| EWC | 2.832 | 0.500 |
| OWM | 2.448 | 0.500 |
| LwF | 3.123 | 0.343 |
| DFWF | 1.849 | 0.689 |
| **RAWM(Ours)** | **0.666** | **0.247** |

(b)

| Method | S | $T_2$ |
|---|---|---|
| Baseline | 0.258 | 46.503 |
| Train-on-All | 0.965 | 2.498 |
| Fine-tune | 8.755 | 5.647 |
| EWC | 3.494 | 5.289 |
| OWM | 3.086 | 5.432 |
| LwF | 4.234 | 4.566 |
| DFWF | 1.874 | 7.355 |
| **RAWM(Ours)** | **1.237** | **3.721** |

(c)

| Method | S | $T_3$ |
|---|---|---|
| Baseline | 0.258 | 91.473 |
| Train-on-All | 1.740 | 0.860 |
| Fine-tune | 20.976 | 2.679 |
| EWC | 5.039 | 2.615 |
| OWM | 4.857 | 2.663 |
| LwF | 2.453 | 2.598 |
| DFWF | 5.324 | 3.275 |
| **RAWM(Ours)** | **2.038** | **1.425** |

Fine-tune, EWC and OWM achieve similar performance in three experiments and the performance of LwF outperforms theirs on the new dataset. In addition, the overcoming forgetting of LwF is also superior to that of the three methods when the feature distributions of new and old datasets are quite different (Table 6c). The DFWF is more effective in overcoming forgetting than the above methods, but its performance on the new dataset is inferior to others. Compared with others, our method achieves lower EER on both old and new datasets of all experiments, which demonstrates that both overcoming forgetting and learning could definitely benefit from our method when training across datasets, regardless of whether the datasets have similar feature distributions (Table 6a, Table 6b) or same languages (Table 6c).

**Sequence training on four datasets**: Finally, We compare our method with several methods for sequence training on four datasets in Table 4b. The results show that the accuracy of DFWF is inferior to others on the last three datasets. which proves that the DFWF is not effective as others in acquiring knowledge. Apart from that, most methods achieve lower EERs than fine-tuning, and the best result for overcoming forgetting and learning is our proposed method, which indicates that the RAWM is superior to others for sequence training on both two and multiple datasets.

## 6 CONCLUSION

In this work, we propose a continual learning algorithm for fake audio detection, called RAWM, that could adaptively modify the weight direction in process of training on new datasets. Our method overcomes catastrophic forgetting by updating weights according to the adaptive modified direction under the restriction of regularization. The experimental results demonstrate that our method performs better than four continual learning methods in learning and overcoming forgetting. The results also prove that our method is effective in overcoming catastrophic forgetting in scenarios of sequence training on both two and multiple datasets. In addition, our method does not require previous data; thus it can be applied to most classification networks. Despite our results, we have yet to study the impact of different acoustic conditions and noise interference on forgetting, and exploring these questions will form the focus of our future studies.

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

# A APPENDIX

## A.1 RESULTS OF OUR BASELINE

Table 7: The EER(%) on multiple evaluation sets. Model-1 to Model-6 are the models trained using the ASVspoof2019LA training set with increasing training steps.

| Model | Evaluation Sets | | | |
|---|---|---|---|---|
| | S | $T_1$ | $T_2$ | $T_3$ |
| Model-1 | 3.751 | **6.316** | **7.670** | **75.198** |
| Model-2 | 2.975 | 8.517 | 10.000 | 78.477 |
| Model-3 | 1.794 | 9.988 | 26.165 | 85.436 |
| Model-4 | **0.258** | 24.532 | 46.503 | 91.473 |
| Model-5 | 0.259 | 25.698 | 44.741 | 91.824 |
| Model-6 | 0.262 | 27.872 | 49.726 | 92.113 |

## A.2 RESULTS OF CONTINUAL LEARNING ON FEW-SHOT

We also present some results of our pre-trained model continually learned on a few samples. In our experiments, only 100 samples randomly selected from new datasets were used for fine-tuning or continual learning. Table 8 shows the results of few-shot continual learning from the **ASVspoof2019** dataset to the **ASVspoof2015** dataset. Most models are trained on the new dataset within five steps. From the results, we can observe that our method RAWM also achieves the best performance on both old and new datasets. By comparing the results in Table 6a and Table 8, we can easily find that reducing the number of samples has only a little damage to our method.

## A.3 CONTINUAL LEARNING FOR SPEECH EMOTION RECOGNITION

Our method can be easily used in other continual learning fields such as image recognition, object detection, and emotion recognition. In the body of the paper, we only demonstrate the application of fake audio detection, which is our current research field.

The key of the RAWM method for other applications is the designation of $\beta$. The most intuitive designation is

$$\beta = \frac{N_1 + N_2 + N_3 + ... + N_k + 1}{N_{(k+1)} + N_{(k+2)} + ... + N_{(k+m)} + 1} \tag{12}$$

Where the $N_i$ represents the number of samples in class $i$. Samples of class 1 - class $k$ are those that have similar feature distributions on old and new datasets (That means, the variance of their feature distributions is small on different datasets). For fake audio detection, the class in the numerator represents genuine audio where $k$ is 1. Apart from that, samples of class $k + 1 -$ class $k + m$ are those that have a great difference in feature distributions. For fake audio detection, the class in the denominator represents fake audio.

For speech emotion recognition, the previous result shows that neutral emotion achieved the highest recognition accuracy across thirteen emotion datasets (Sharma, 2022). So we infer that neutral speech has a more similar feature distribution than that of happy, sad, and angry, thus the ratio hyperparameter $\beta$ of our method can be written as follows.

$$\beta = \frac{N_{neutral} + 1}{N_{happy} + N_{angry} + N_{sad} + 1} \tag{13}$$

Based on this inference, we also conduct some experiments for speech emotion recognition. We choose four emotional classes, including neutral, happy, angry, and sad. The results have been added to Table 9 It could be easily observed that our method still achieves the highest accuracy on both datasets compared with other continual learning methods.

Table 8: The EER(%) of few-shot experiments. All experiments are first trained using the training set of **ASVspoof2019** and then trained on the subset of the training set of **ASVspoof2015**. The subset only includes 100 samples randomly chosen from the training set of **ASVspoof2015**. All experiments are evaluated using the evaluation set on **ASVspoof2019** and **ASVspoof2015**.

| Method | ASVspoof2019 | ASVspoof2015 |
|---|---|---|
| Baseline | 0.258 | 24.532 |
| Fine-tune | 7.951 | 0.617 |
| EWC | 2.972 | 0.619 |
| OWM | 2.683 | 0.617 |
| LwF | 3.198 | 0.542 |
| DFWF | 1.975 | 0.733 |
| **RAWM(Ours)** | **0.923** | **0.312** |

Table 9: The Acc(%) of various continual learning methods for 4-classes speech emotion recognition. All experiments are trained using the training set in order to $\mathbf{MSP-Podcast} \rightarrow \mathbf{IEMOCAP}$ and are evaluated using the evaluation set on $\mathbf{MSP-Podcast}$ and **IEMOCAP**

| Method | MSP − Podcast | IEMOCAP |
|---|---|---|
| Baseline | 54.446 | 30.043 |
| Fine-tune | 24.094 | 50.379 |
| OWM | 32.267 | 50.162 |
| LwF | 38.800 | 44.034 |
| **RAWM(Ours)** | **41.995** | **54.229** |

