# OpenReview forum: "Do You Remember? Overcoming Catastrophic Forgetting for Fake Audio Detection"
_ICLR.cc/2023/Conference — Submitted to ICLR 2023_

### Official Review · Reviewer_VFqH · 2022-10-22

**Confidence:** 4
**Correctness:** 3
**Technical Novelty And Significance:** 2
**Empirical Novelty And Significance:** 2
**Recommendation:** 5

**Clarity, Quality, Novelty And Reproducibility:**

This paper presents the problem quite clearly, however, the reason for using projector Q instead of normal gradient need to be explained.

**Strength And Weaknesses:**

Strength:
The proposed continual learning algorithm is designed for the application of fake audio detection. The authors observe some problems when using conventional continual learning algorithms and propose two methods (adaptive modification direction and a regularization constraint) to overcome them.

Weaknesses:
To avoid using training data from the previous tasks, the proposed regularization constraint is based on a teacher model, which is not a very novel idea. On the other hand, the way to apply an adaptive modification direction is not very clear to me. In page 4, the authors claim that “In the training process on the new dataset, they should be trained without any modification.” However, the authors propose a new projector Q which is orthogonal to the projector P from OWM. My question is: why not just use the original gradient from SGD, ∆W^BP in this case?
Other comments:
1. Is P a square matrix? Otherwise, how to compute Q using e.q. (4)?

2. In the original paper of ‘OWM’ (Continual Learning of Context-dependent Processing in Neural Networks), the authors claim that ” Recursive Least Square (RLS) algorithm which can be used to train feedforward and recurrent neural networks to achieve fast convergence’ However in this paper, the proposed fake audio detection model is based on Wav2vec (which includes convolutional encoder) and a CNN classifier. I’m not sure how OWM performs on convolutional layers. My concern is that I think OWM assumes the calculation between input and model weights is through matrix multiplication instead of convolutional operation.

3. In Fig. 1. of the original paper of ‘OWM’, they claim that OWM can reach the position inside the overlapping subspace between two tasks. However, in Fig. 1. of your paper, you mention that OWM can not reach the region of dataset 2. Please explain why there is such a difference.

4. In Table 6, why even Fine-tune cannot obtain the best performance on the new task compared to other methods?

5. It would be great to also report the ‘experience-replay-based’ method in the experiment.

Typo:
1. Page 1, fae audio detection -> fake audio detection

2. Page 3, yo and yo are the old and new ground truth -> yo and yn are the old and new ground truth


**Summary Of The Paper:**

This paper proposes a continual learning algorithm, called Regularized Adaptive Weight Modification (RAWM) to overcome catastrophic forgetting for fake audio detection. RAWM is based on the previously published orthogonal weight modification (OWM). Because OWM does not consider the similarity of some audio, including fake audio obtained by the same algorithm and real audio, on different datasets. To solve this limitation, adaptive modification direction and a regularization constraint are proposed. Experimental results show a good performance improvement compared to baselines.

**Summary Of The Review:**

This paper is based on the previously published OWM paper, and two modifications (adaptive modification direction and a regularization constraint) are proposed. As pointed out earlier, the reason to apply Q in adaptive modification direction is unclear, and the novelty of using a teacher model for regularization constraint is somewhat limited.

---

> ### Author Response · Authors · 2022-11-19
> **Response to Reviewer VFqH**
>
> $Q_1$: To avoid using training data from the previous tasks, the proposed regularization constraint is based on a teacher model, which is not a very novel idea. On the other hand, the way to apply an adaptive modification direction is not very clear to me. In page 4, the authors claim that “In the training process on the new dataset, they should be trained without any modification.” However, the authors propose a new projector Q which is orthogonal to the projector P from OWM. My question is: why not just use the original gradient from SGD, ∆W^BP in this case?
>
> $A_1$: Thanks for your advice. In this paper, we aim to overcome catastrophic forgetting while exerting a positive influence on acquiring new knowledge without any previous samples.
>
> As for your question. The “they” mentioned in the statement “In the training process on the new dataset, they should be trained without any modification.” represents the genuine audio in every training batch. As we all know, there are genuine audio and fake audio in a batch as usual, so we introduce the projector Q which is orthogonal to the projector P to modify the weight direction R according to the ratio of genuine and fake audio automatically ( The more the number of genuine audio, the greater the norm of projector Q, that makes the modification direction R closer to the space of previous data. ). In the above statement, we express that we should train genuine audio without any modification, which is based on the ideal genuine audio which has a very similar acoustic feature distribution on different datasets, and we can train models as you advised. But in practice, most of genuine audios still have a small or great difference in their acoustic feature distribution (e.g., ASVspoof2019 to VCC, which is English to Multilingual; ASVspoof2019 to In-the-Wild, which is audio from laboratory to audio from real-world), thus we introduce two measures to solve this problem: 1): We introduce a projector Q to computing the weight direction R rather than just using the original gradient from SGD.(Sec 4.1) 2): We introduce a regularization (See 4.2)
>
> $Q_2$: Is P a square matrix? Otherwise, how to compute Q using e.q. (4)?
>
> $A_2$: Yes.we also added this explanation to our paper.
>
> $Q_3$: I’m not sure how OWM performs on convolutional layers.
>
> $A_3$: Yes, it is a track in the official code of the OWM to solve this problem. The authors of the original OWM paper release their code, which computes the P in convolutional layer by decomposing the mean output $\bar{x}$ into some parts and then calculating them as the process of linear layers. The details can be seen on official code of OWM or our code.
>
> $Q_4$: In Fig. 1. of the original paper of ‘OWM’, they claim that OWM can reach the position inside the overlapping subspace between two tasks. However, in Fig. 1. of your paper, you mention that OWM can not reach the region of dataset 2. Please explain why there is such a difference.
>
> $A_4$: The overlapping space in our paper is not as same as that mentioned in the original OWM paper, which can be viewed as the subspace of the overlapping space in the OWM paper. This Fig and statement are based on our experimental results. The OWM authors only compared the OWM with the SGD. Their experimental results show that the OWM can achieve great performance on new dataset as the SGD and overcome forgetting on old dataset, so they claim that OWM can reach the overlapping space. But in our paper, the experimental results in Table 6 demonstrate that our method has achieved better performance on new dataset that the OWM. So, we narrowed the overlapping space and redefined the “Great Performance” on dataset 2. In this situation, the OWM can not reach the region of new dataset.
>
> $Q_5$: In Table 6, why even Fine-tune cannot obtain the best performance on the new task compared to other methods?
>
> $A_5$: Yes. Please see the explanation in "Response to results in Table. 6"
>
> $Q_6$: It would be great to also report the ‘experience-replay-based’ method in the experiment.
>
> $A_6$: Yes. Thanks for your advice. In addition, we present the results of training all datasets (Tain-on-All), which is worth noting that their EER is the lower bound to all continual-learning-based methods we mentioned.
>
> $Q_7$: Typo:
> 1.	Page 1, fae audio detection -> fake audio detection
> 2.	Page 3, yo and yo are the old and new ground truth -> yo and yn are the old and new ground truth
>
> $A_7$: 36.	Yes. Thanks. We have revised accordingly.

---

> > ### Author Response · Authors · 2022-11-19
> > **Response to results in Table. 6**
> >
> > Thanks for your question.
> >
> > In continual learning research, a good continual learning method aims to decrease the forgetting of learned knowledge without skewing the learning performance on the target dataset. For example, the EWC method consolidates the model weights that are crucial to learning knowledge, if these important weights are close to the best weights trained on the target dataset, the consolidation of the EWC will beneficial to training on the target dataset, thus outperforming than fine-tuning.
> >
> > In fact, the results in Table 6 are averaged values over 7 runs and only the LwF and our method obviously outperform fine-tuning, the results on target datasets of EWC and OWM are very close to fine-tuning. In fact, among our experimental results, the EER of the EWC and OWM is marginally higher than that of fine-tuning sometimes (2, or 3 runs), but they are marginally lower in most situations, so the averaged results of the EWC and OWM are very close to fine-tuning. For the LwF, we conducted many experiments to search a most suitable \lambda_0 (mentioned in Sec 3.2), thus achieving obvious improvement. The original paper of LwF (Ref. Li & Hoiem, 2017.) also reports the same results. For our method, we modify the weight direction close to an orthogonal projector to the previous data’s subspace to protect learning on new fake audio because the knowledge of previous fake audio will damage the learning performance on target dataset (Their acoustic feature are quietly different), and we also modify the weight direction close to previous subspace when the number of genuine audio is bigger than that of fake audio, like experience-replay-based method, thus improving the learning performance on target dataset.

---

### Official Review · Reviewer_jUDi · 2022-10-24

**Confidence:** 3
**Correctness:** 2
**Technical Novelty And Significance:** 3
**Empirical Novelty And Significance:** 2
**Recommendation:** 3

**Clarity, Quality, Novelty And Reproducibility:**

**Clarity**

Overall, I found this paper to be quite unclear and required several passes to understand. A primary issue is definition and consistency of terminology. For example, this paper primarily concerns “fake audio detection” and “continual learning” but neither are defined anywhere. Also, this paper fluctuates between using “source / target” (in Figure 1 / Experiments) and “old / new” (throughout) naming conventions for adaptation. Important notations like T_1, T_2, and T_3 are introduced in the header of Table 1 instead of Section 5.1 where the reader might expect to see them.


A key question I was left with after reading this paper: in Table 6, why do the majority of the continual learning algorithms outperform simple fine-tuning on the target dataset? This may be an understood phenomenon in the continual learning research community but I find it incredibly counterintuitive as an outsider. Can the authors clarify the “free lunch” aspect of continual learning, especially in this context of fake audio detection?

Another comment is that the introduction could do more to clarify the specific assumptions about fake audio detection that motivate the need for tailored continual learning algorithms. The third paragraph of the intro attempts to do this in one sentence (starting with “Because the…”), but this sentence is exceptionally difficult to parse and entangled w/ the proposed method (instead of stated in a method-agnostic fashion). A clearer upfront explanation of this would have greatly improved my ability to understand the rest of the paper on the first pass.

There are also numerous places throughout the paper containing misleading or ambiguous statements. For example, the first paragraph of section 5.1 suggests that the training subset of the source dataset will be used, but the training subsets of the target datasets will not be (which is wrong in the context of continual learning).

** Quality **: Accepting the sole focus on continual learning, the experiments are reasonably well-designed. However, I would still have very much liked to see a comparison to “oracle” methods w/ access to the source dataset - how much performance do we lose by making the assumption that source datasets are inaccessible?

** Novelty **: The proposed method is adequately interesting and novel - a stronger version of this paper might explore the proposed method for other tasks (besides fake audio detection) with similar structure.

** Reproducibility **: It would be an extraordinary challenge to reproduce this algorithm from the notation / information in this paper alone - would the authors be able to release code?


**Strength And Weaknesses:**

The primary strength of this paper is that of the results: the proposed algorithm clearly outperforms other comparable continual learning algorithms for standard fake audio detection tasks. The primary weakness is a lack of motivation for the problem formulation explored here. Specifically, this paper explores continual learning for fake audio detection, for which there are two key issues:

(1) I would argue that the biggest practical challenge in fake audio detection is not _adapting_ to new domains w/ many examples (as is explored in continual learning) but rather improving _zero/few-shot generalization_ to new domains. The threat model in fake audio detection should emphasize the persistent development of new methods for generating fake audio. Instead, the exploration of continual learning suggests a perpetual cat-and-mouse game where detection models must be manually adapted to new domains.

(2) The motivation for the _continual_ learning setup (where source data is inaccessible) is unclear. The introduction says in one sentence that “in some practical situations, it is almost impossible to obtain the old data”, but does not suggest any particular situations where that would be the case. Moreover, all of the experiments _contrive_ such situations (since the source dataset is always accessible), and there are no comparisons to “oracle” performance w/ access to source data.

Additionally, this paper struggles w/ clarity (see next section).

**Summary Of The Paper:**

This paper proposes a new method for continual learning which leverages structure found in the particular task of fake audio detection. Specifically, the proposed method augments continual learning algorithms by leveraging structural differences found in fake audio detection between two individual distribution shifts latent in the source and target datasets: source fake -> target fake, and source real -> target real. Compared to other continual learning algorithms, the proposed algorithm leads to large improvements in performance on both source and target datasets.

**Summary Of The Review:**

Overall, this paper presents some interesting ideas and compelling results, but the current version struggles w/ (1) motivating the setting of interest (continual learning for fake audio detection) including lack of comparison to methods which can access the source data, and (2) clarity of explanation.

---

> ### Author Response · Authors · 2022-11-19
> **Response to Reviewer jUDi**
>
> $Q_1$:  I would argue that the biggest practical challenge in fake audio detection is not adapting to new domains w/ many examples (as is explored in continual learning) but rather improving zero/few-shot generalization to new domains. The threat model in fake audio detection should emphasize the persistent development of new methods for generating fake audio.
>
> $A_1$: Thanks for your advice and I think it is a great contribution to our work in the foreseeable future. There are two points we would like to express.
>
> The first is that it is also a viable option for a few samples to be trained using continual learning methods. Because the pretrain – finetune framework has been widely used in continual learning, which proves feasible for few-shot learning, most methods, especially those we mentioned in this paper, under this framework have promising performance on zero/few-shot generalization to new domains. In addition, we also present some experimental results that we trained the model in 100 samples with different continual learning methods, which has added and highlighted in Appendix. 2 and Table. 8.
>
> The second is that it seems too ambitious for fake audio detection to adaptively deal with those fake audio generated by new methods. The continual learning method, however, is one of the promising methods to address this issue in this regard.
>
> $Q_2$: The motivation for the continual learning setup (where source data is inaccessible) is unclear. The introduction says in one sentence that “in some practical situations, it is almost impossible to obtain the old data”, but does not suggest any particular situations where that would be the case. Moreover, all of the experiments contrive such situations (since the source dataset is always accessible), and there are no comparisons to “oracle” performance w/ access to source data.
>
> $A_2$: Thanks for your advice. The motivation has been highlighted in the introduction. There are many situations where we can not access source data. For instance, a pre-trained model proposed by a company has been released to the public. It is unfeasible for the public to fine-tune it using the data belonging to the original company. var We have added this situation’s description to our paper in the second paragraph of the introduction and highlighted it.
>
> We have also conducted some experiments in which we trained our models on all training sets, the results have been added and highlighted in Table 4 and Table 6.
>
> $Q_3$: Overall, I found this paper to be quite unclear and required several passes to understand. A primary issue is definition and consistency of terminology. For example, this paper primarily concerns “fake audio detection” and “continual learning” but neither are defined anywhere. Also, this paper fluctuates between using “source / target” (in Figure 1 / Experiments) and “old / new” (throughout) naming conventions for adaptation. Important notations like T_1, T_2, and T_3 are introduced in the header of Table 1 instead of Section 5.1 where the reader might expect to see them.
> Another comment is that the introduction could do more to clarify the specific assumptions about fake audio detection that motivate the need for tailored continual learning algorithms.
> There are also numerous places throughout the paper containing misleading or ambiguous statements. For example, the first paragraph of section 5.1 suggests that the training subset of the source dataset will be used, but the training subsets of the target datasets will not be (which is wrong in the context of continual learning).
>
> $A_3$: Thanks. We have revised them accordingly.
>
> First of all, we apologize for those difficult understanding descriptions. We have now rewritten some of the introduction, especially the third paragraph, and highlighted the motivation and our method explanation so as to make it read easier. All statements “source/target” have been replaced by “old/new”. The notations like T_1, T_2 and T_3 you mentioned have been introduced in Sec 5.1 rather than the Table, the misunderstanding description in Sec 5.1 is also revised and highlighted.
>
> Here we give the definitions of continual learning and fake audio detection.
> Continual Learning: that is, the ability to learn consecutive tasks without forgetting how to perform previously trained tasks.
> Fake Audio Detection: that is, the ability to detect fake audio from a collection of mixed genuine and fake audio.
>
> $Q_4$: A key question I was left with after reading this paper: in Table 6, why do the majority of the continual learning algorithms outperform simple fine-tuning on the target dataset?
>
> $A_4$: Yes. Please see our response "Response to results in Table. 6".
>
> $Q_5$:  It would be an extraordinary challenge to reproduce this algorithm from the notation / information in this paper alone - would the authors be able to release code?
>
> $A_5$: Yes. The code will be publicly available in the foreseeable future.

---

> > ### Author Response · Authors · 2022-11-19
> > **Response to results in Table. 6**
> >
> > Thanks for your question.
> >
> > In continual learning research, a good continual learning method aims to decrease the forgetting of learned knowledge without skewing the learning performance on the target dataset. For example, the EWC method consolidates the model weights that are crucial to learning knowledge, if these important weights are close to the best weights trained on the target dataset, the consolidation of the EWC will beneficial to training on the target dataset, thus outperforming than fine-tuning.
> >
> > In fact, the results in Table 6 are averaged values over 7 runs and only the LwF and our method obviously outperform fine-tuning, the results on target datasets of EWC and OWM are very close to fine-tuning. In fact, among our experimental results, the EER of the EWC and OWM is marginally higher than that of fine-tuning sometimes (2, or 3 runs), but they are marginally lower in most situations, so the averaged results of the EWC and OWM are very close to fine-tuning. For the LwF, we conducted many experiments to search a most suitable \lambda_0 (mentioned in Sec 3.2), thus achieving obvious improvement. The original paper of LwF (Ref. Li & Hoiem, 2017.) also reports the same results. For our method, we modify the weight direction close to an orthogonal projector to the previous data’s subspace to protect learning on new fake audio because the knowledge of previous fake audio will damage the learning performance on target dataset (Their acoustic feature are quietly different), and we also modify the weight direction close to previous subspace when the number of genuine audio is bigger than that of fake audio, like experience-replay-based method, thus improving the learning performance on target dataset.

---

### Official Review · Reviewer_d2it · 2022-10-27

**Confidence:** 3
**Clarity, Quality, Novelty And Reproducibility:** 1. Clear Enough
2. Good Quality
3. I …
**Correctness:** 4
**Technical Novelty And Significance:** 2
**Empirical Novelty And Significance:** 2
**Recommendation:** 6

**Strength And Weaknesses:**

Strengths:
1. Paper is well written and the solution proposed addresses the problem quite well.
2. The experimental validation is sufficiently elaborate and shows good improvement over the baselines considered.

Weakness:
1. There are a few parts of the paper which are not particularly well written and makes it a difficult read. For instance:
 a) In Section 4.2 the notations and equations 9, 10 are not in sync.
 b) Symbols and notations are not clearly explained on several occasions.
 c) Out of nowhere in Section 5.3, authors start talking about a "Model-4
2. How does the proposed method compare to training on all datasets ?
3. It is not clear why the assumption that genuine audio across different datasets is more likely to be similar than fake audios from different datasets is necessary ? Can the authors not think about exploiting fake audio similarity as well in their technique ?

**Summary Of The Paper:**

The authors address the problem of "catastrophic forgetting" in the context of fake audio detection. When a network trained on one dataset (D1) and is fine tuned on another dataset (D2), the fine-tuned network loses its original performance on the D1. This is referred to as catastrophic forgetting for the considered task of fake audio detection. There are several approaches addressing this problem. One of the recent approaches is using the orthogonal weight modification (OWM) technique. This is based on the viewpoint that modifying the weights in a direction orthogonal to the Weights subspace corresponding to D1 would ensure that the modified network's performance on D1 would not deteriorate. Thus, the idea while fine-tuning on D2 is to modify the weight update equation by modifying the weights in a direction orthogonal to that corresponding to weights subspace representing D1. Such a weight modification has a regularization effect on the network to retain its performance on D1. This however presents a significant drawback: The Performance of this regularized network on D2 suffers compared to without regularization.

In this paper the authors address the above two issues in OWM by proposing Regularized Adaptive Weight Modification (RAWM) where the above two problems are addressed respectively by a nice observation that blindly updating the weights in an orthogonal direction to the subspace corresponding to D1 might be a bad idea as there could be several genuine audios in D1 which are similar to D2 and for those samples - the regular update might not harm network performance on D2 while improving its performance on D2. This is accounted in the weight update by modifying the direction of update to be based on the number of genuine vs fake examples in each batch and using this to adapt the orthogonal weight update. In the degenrate case of all fake examples this proposed approach would boil down to the OWM method.

Overall the paper is well written with clear problem definition, experimental results and useful comparisons.

**Summary Of The Review:**

Overall the proposed solution of adapting the weight modification by accounting for the genuine and fake samples in each batch seems interesting and shows promising results. The methodology is clearly explained with detailed insightful evaluation.

---

> ### Author Response · Authors · 2022-11-19
> **Response to Reviewer d2it**
>
> $Q_1$:  There are a few parts of the paper which are not particularly well written and makes it a difficult read. For instance: a) In Section 4.2 the notations and equations 9, 10 are not in sync. b) Symbols and notations are not clearly explained on several occasions. c) Out of nowhere in Section 5.3, authors start talking about a "Model-4"
>
> $A_1$: Thanks for your correction. We have added some explanations and rewritten some descriptions in Sec 4 to make it read easier, especially in Sec 4.2 and Sec 4.3. As for the confusion “Model-4” you mentioned in Sec 5.3, it represents our baseline in Table. 7 originally and has been replaced by “our baseline”.
>
> $Q_2$:  How does the proposed method compare to training on all datasets?
>
> $A_2$: We have added the result of training on all datasets in Table 4 and Table 6. Thanks for your advice.
>
> $Q_3$: It is not clear why the assumption that genuine audio across different datasets is more likely to be similar than fake audio from different datasets is necessary? Can the authors not think about exploiting fake audio similarity as well in their technique?
>
> $A_3$: Thanks for your questions. Some papers referred by us (Todisco et al., 2019; Ma et al., 2021; Yan et al., 2022) show that genuine audio across different datasets is more likely to be similar than fake audio. This assumption is essential for our method in computing ratio hyperparameter $\beta$ which affects the modified direction. We show the details in Sec 4.1.
>
> For the second question, we also consider that some fake audio may have similar features on different datasets. However, for lacking fake audio’s fingerprint (Fingerprint is a sign of fake audio, which often represents the type of voice conversion algorithm or speech synthesis algorithm. Most of the fake audio datasets and real-world applications have not provided fake types of fake audio.),  it is difficult to recognize their fingerprint thus impossible to compute a ratio as β to modify weight direction. So, we introduce a regularization (see Sec 4.2) to protect the learned knowledge of this part of fake audio and the genuine audios that have unsimilar features on different datasets.
>
> $Q_4$:  I am not entirely sure this work is reproducible.
>
> $A_4$: Yes. The code will be publicly available in the foreseeable future.

---

### Official Review · Reviewer_a37q · 2022-11-01

**Confidence:** 3
**Correctness:** 3
**Technical Novelty And Significance:** 3
**Empirical Novelty And Significance:** 3
**Recommendation:** 6

**Clarity, Quality, Novelty And Reproducibility:**

Clarity: Overall the paper is clear. However, some of the presentation can be made more clear (see Weaknesses) and there are also a few typos in the paper (see Minor comments).

Quality: The proposed method is well-motivated and experiments demonstrate the effectiveness of the proposed method. However, the proposed method is limited to the task of fake audio detection.

Novelty: The proposed method is incremental to OWM, but provides additional novel technical insights for the problem of fake audio detection.

Reproducibility: The paper seems reproducible.


**Strength And Weaknesses:**

Strength

The paper proposes a novel method called Regularized Adaptive Weight Modification (RAWM) to address the problem of catastrophic forgetting in the context of fake audio detection. The proposed method builds on prior work orthogonal weight modification (OWM).

The approach is well-motivated (genuine audios are more similar than fake audios in different datasets. The proposed method obtains a better tradeoff between learning from a new dataset while not forgetting the past knowledge.)

The paper provides a satisfactory literature review.

The paper evaluates the proposed approach across multiple fake audio detection datasets and shows promising results in the continual learning setting.

Weaknesses

The proposed method seems to be general for continual learning. However, experiments only demonstrate its value for the task of fake audio detection. Can authors comment on whether the proposed method can benefit any other tasks other than fake audio detection? The paper can be made stronger if the proposed method can be extended to more applications or to general recognition tasks (e.g., image/video/audio recognition).

Does the proposed method also work for multiclass classification tasks? If so, what are the necessary changes?

Presentation of section 3 can be improved due to lack of context. It will be useful to include a paragraph to describe the problem setting (e.g., notation of model layers, notation for each dataset, notation for gradient etc.). A lot of the context is only introduced in section 4 (“We consider a feed-forward network consisting of L + 1 layers..”).


Minor comments:

In the second equation of EQ1,  the numerator is P_l(i-1,j) \bar{X}(l-1). Should \bar{X}(l-1) be indexed with i,j ?

Section 3.2, “y_o and y_o are the old and new ground truth” (typos?).




**Summary Of The Paper:**

The paper considers the problem of continual learning in the context of fake audio detection. One of the main challenges of continual learning is catastrophic forgetting. The paper proposes a new algorithm called Regularized Adaptive Weight Modification (RAWM). The motivation is genuine audios are more similar than fake audios in different datasets. The proposed approach can adaptively modify the direction of weights according to the ratio of genuine audio and fake audio of each batch in the process of fine-tuning. The paper evaluated the proposed approach across multiple fake audio detection datasets and showed promising results in the continual learning setting.


**Summary Of The Review:**

The paper proposes a novel method (RAWM) for continual learning in the context of fake audio detection. Experiments demonstrate that the proposed method obtains strong performance compared to the prior methods. The paper can be made stronger if the proposed method can be extended to more applications or to general recognition tasks.

---

> ### Author Response · Authors · 2022-11-19
> **Response to Reviewer a37q**
>
> $Q_1$: The proposed method seems to be general for continual learning. However, experiments only demonstrate its value for the task of fake audio detection. Can authors comment on whether the proposed method can benefit any other tasks other than fake audio detection? The paper can be made stronger if the proposed method can be extended to more applications or to general recognition tasks (e.g., image/video/audio recognition).
>
> $A_1$:
> Yes, our method can be easily used in other continual learning fields such as image recognition, object detection, and emotion recognition. Because of the limitation of the paper s length, we only demonstrate the application of fake audio detection, which is our current research field.
>
> The key of the RAWM method for other applications is the designation of $\beta$. The most intuitive designation is
> \begin{equation}
> \beta=\frac{N_1+N_2+N_3+...+N_k+1}{N_(k+1)+N_(k+2)+...+N_(k+m)+1}
> \end{equation}
> Where the $N_i$ represents the number of samples in class i. Samples of $class_{1 - k}$ are those that have similar feature distributions on source and target datasets (That means, the variance of their feature distributions is small on different datasets). For fake audio detection, the class in the numerator represents genuine audio where k is 1. Apart from that, samples of $class_{k+1 – k+m}$ are those that have a great difference in feature distributions on source and target datasets. For fake audio detection, the class in denominator represents fake audio.
>
> For speech emotion recognition, the previous result shows that neutral emotion achieved the highest recognition accuracy across thirteen emotion datasets. So we infer that neutral speech has a more similar feature distribution than that of happy, sad, and angry, thus the $\beta$ be written as follows.
> \begin{equation}
> \beta=\frac{N_{neu}+1}{N_{hap}+N_{ang}+N_{sad}+1}
> \end{equation}
> We also conduct some experiments on speech emotion recognition, the results have been added to Appendix A.2.
>
> (PS: In MULTI-LINGUAL MULTI-TASK SPEECH EMOTION RECOGNITION USING WAV2VEC 2.0, the model achieves the highest accuracy on neutral emotion than others. All experiments are conducted on 13 multi-lingual datasets and shared with the same parameters. The results, to some extent, indicate that neutral emotion has a more similar feature distribution than others.)
>
> $Q_2$: Does the proposed method also work for multiclass classification tasks? If so, what are the necessary changes?
>
> $A_2$: Yes. Please see Answer 1.
>
> $Q_3$: Presentation of section 3 can be improved due to lack of context. It will be useful to include a paragraph to describe the problem setting (e.g., notation of model layers, notation for each dataset, notation for gradient etc.). A lot of the context is only introduced in section 4 (“We consider a feed-forward network consisting of L + 1 layers..”)
>
> $A_3$: Yes, thanks for your advice. We have added and highlighted a paragraph to explain the problem setting in Sec. 3.1.
>
> $Q_4$: In the second equation of EQ1, the numerator is P_l(i-1,j) \bar{X}(l-1). Should \bar{X}(l-1) be indexed with i,j ?
> Section 3.2, “y_o and y_o are the old and new ground truth” (typos?).
>
> $A_4$: Yes, thanks for your correction. The EQ you mentioned have been corrected in Sec. 3.1; As for Sec.3.2, we have rewritten some descriptions and added some explanations to make it read easier. All of them have been highlighted in Sec.3.2.

---

### Author Response · Authors · 2022-11-19
**Response to All Reviews**

Dear reviewers:

We would like to, first and foremost, present our deepest gratitude for your time and in reading our manuscript, as well as for the comments and critiques. We have modified our manuscript accordingly (significant revisions have been highlighted). Please find our responses to your questions, concerns, and critiques below.

---

### Decision · Program_Chairs · 2023-01-20

**Decision:**

Reject

**Justification For Why Not Higher Score:**

Even in the revised version of the paper, a number of issues remain that should be addressed before the paper can be accepted:
- Authors argue that length restricts them from demonstrating the method on more tasks, yet they add a description to the appendix. It should be possible to do so in the main text, and maybe readers would find a more standard task than multi-modal, multi-lingual emotion recognition more intuitive to understand
- Reviewers are still wondering why not just use the original gradient from SGD, ∆W^BP in this case (also orthogonal to the projector P)?
- Results shown in Table 6 regarding fine-tuning on the new tasks need to be clarified and discussed further.


**Justification For Why Not Lower Score:**

n/a

**Metareview: Summary, Strengths And Weaknesses:**

Summary

This paper proposes "Regularized Adaptive Weight Modification" (RAWM), a continual learning algorithm to overcome catastrophic forgetting for fake audio detection. RAWM is based on the previously published orthogonal weight modification (OWM). OWM does not consider the similarity of some audio classes, e.g. fake audio obtained by the same algorithm or recorded audio, across different datasets. To solve this limitation, authors propose adaptive modification direction (based on the prevalence of said classes) and a regularization constraint. Experimental results show performance improvements compared to baselines.

Strengths

- RAWM is an incremental improvement to address the problem of catastrophic forgetting. The proposed method builds on prior work on orthogonal weight modification (OWM).
- The approach is well-motivated (genuine audios are more similar than fake audios in different datasets. The proposed method obtains a better tradeoff between learning from a new dataset while not forgetting the past knowledge.)
- The experimental section is carefully done, evaluating on multiple fake audio detection datasets.
- The paper is well written and contains a good literature overview, the proposed approach is well motivated

Weaknesses

- The proposed method seems general, but the paper does not apply it to any other tasks. Additional tasks would make the paper stronger, since the method would be shown to generalize.
- In a similar vein: Does the proposed method also work for multiclass classification tasks? If so, what are the necessary changes? These two questions have been addressed to some extent in the revised version, but it would be better to do so in the main text (rather than the appendix) and with a more standard experimental setup than currently done.
- Several sections' (e.g. 3, 4) presentation can be improved: consistency of notation, missing definitions, etc
- Basic ablations (how does the proposed method compare to training/ fine-tuning on all datasets) are missing
- The use case of fake audio detection using continual learning is likely unfamiliar to many readers, despite the authors' elaboration.


**Summary Of Ac-Reviewer Meeting:**

n/a